# How Many Samples are Needed to Estimate a Convolutional Neural Network?

**Simon S. Du***
Carnegie Mellon University

**Yining Wang***
Carnegie Mellon University

**Xiyu Zhai**
Massachusetts Institute of Technology

**Sivaraman Balakrishnan**
Carnegie Mellon University

**Ruslan Salakhutdinov**
Carnegie Mellon University

**Aarti Singh**
Carnegie Mellon University

## Abstract

A widespread folklore for explaining the success of Convolutional Neural Networks (CNNs) is that CNNs use a more compact representation than the Fully-connected Neural Network (FNN) and thus require fewer training samples to accurately estimate their parameters. We initiate the study of rigorously characterizing the sample complexity of estimating CNNs. We show that for an $m$-dimensional convolutional filter with linear activation acting on a $d$-dimensional input, the sample complexity of achieving population prediction error of $\epsilon$ is $\widetilde{O}(m/\epsilon^2)$ [2], whereas the sample-complexity for its FNN counterpart is lower bounded by $\Omega(d/\epsilon^2)$ samples. Since, in typical settings $m \ll d$, this result demonstrates the advantage of using a CNN. We further consider the sample complexity of estimating a one-hidden-layer CNN with linear activation where both the $m$-dimensional convolutional filter and the $r$-dimensional output weights are unknown. For this model, we show that the sample complexity is $\widetilde{O}\left((m+r)/\epsilon^2\right)$ when the ratio between the stride size and the filter size is a constant. For both models, we also present lower bounds showing our sample complexities are tight up to logarithmic factors. Our main tools for deriving these results are a localized empirical process analysis and a new lemma characterizing the convolutional structure. We believe that these tools may inspire further developments in understanding CNNs.

## 1 Introduction

Convolutional Neural Networks (CNNs) have achieved remarkable impact in many machine learning applications, including computer vision (Krizhevsky et al., 2012), natural language processing (Yu et al., 2018) and reinforcement learning (Silver et al., 2016). The key building block of these improvements is the use of convolutional (weight sharing) layers to replace traditional fully connected layers, dating back to LeCun et al. (1995). A common folklore of explaining the success of CNNs is that they are a more compact representation than Fully-connected Neural Networks (FNNs) and thus require fewer samples to estimate. However, to our knowledge, there is no rigorous characterization of the sample complexity of learning a CNN.

The main difficulty lies in the convolution structure. Consider the simplest CNN, a single convolutional filter with linear activation followed by average pooling (see Figure 1a), which represents a

function $F_1 : \mathbb{R}^d \mapsto \mathbb{R}$ of the form:

$$F_1(x; w) = \sum_{\ell=0}^{r-1} w^\top \mathsf{P}_s^\ell x, \tag{1}$$

where $w \in \mathbb{R}^m$ is the filter of size $m$ and a stride size of $s$, $r \approx d/s$ is the total number of times filter $w$ is applied to an input vector $x \in \mathbb{R}^d$, and $\mathsf{P}_s^\ell x := [x_{\ell s+1}, x_{\ell s+2}, \ldots, x_{\ell s+m}]$ is an $m$-dimensional segment of the feature vector $x$. Noting that $F_1$ is a linear function of $x$, we can also represent $F_1$ by a one-layer fully connected neural network (linear predictor):

$$F_1^{\mathrm{FNN}}(x, \theta) = \theta^\top x \tag{2}$$

for some $\theta \in \mathbb{R}^d$. Suppose we have $n$ samples $\{x_i, y_i\}_{i=1}^n$ where $x$ is the input and $y$ is the label and use the least squares estimator:

$$\widehat{\theta} := \arg\min_{\theta \in \mathbb{R}^d} \sum_{i=1}^n (y_i - \theta^\top x_i)^2.$$

By a classical results analyzing the prediction error for linear regression (see for instance (Wasserman, 2013)), under mild regularity conditions, we need $n \asymp d/\epsilon^2$ to have $\sqrt{\mathbb{E}_{x \sim \mu} |\widehat{\theta}^\top x - \theta_0^\top x|^2} \leqslant \epsilon$, where $\mu$ is the input distribution and $\theta_0$ is the optimal linear predictor. The proof for FNN is fairly simple because we can write $\widehat{\theta} = \left( X^\top X \right)^{-1} X^\top Y$ (normal equation) where $X$ and $Y$ are the aggregated features and labels, respectively and then directly analyze this expression.

On the other hand, the network $F_1$ can be viewed as a linear regression model with respect to $w$, by considering a "stacked" version of feature vectors $\widetilde{x}_i = \sum_{\ell=0}^{r-1} \mathsf{P}_s^\ell x \in \mathbb{R}^m$. The classical analysis of ordinary least squares in linear regression does not directly yield the optimal sample complexity in this case, because the distributional properties of $\widetilde{x}_i$ as well as the spectral properties of the sample covariance $\sum_i \widetilde{x}_i \widetilde{x}_i^\top$ are difficult to analyze due to the heavy correlation between coordinates of $\widetilde{x}$ corresponding to overlapping patches. We discuss further details of this aspect after our main positive result in Theorem 1.

In this paper, we take a step towards understanding the statistical behavior of the CNN model described above. We adopt tools from localized empirical process theory (van de Geer, 2000) and combine them with a structural property of convolutional filters (see Lemma 2) to give a complete characterization of the statistical behavior of this simple CNN.

We first consider the problem of learning a convolutional filter with average pooling as in Eq.(1) using the least squares estimator. We show in the standard statistical learning setting, under fairly natural conditions on the input distribution, $\widehat{w}$ satisfies

$$\sqrt{\mathbb{E}_{x \sim \mu} |F_1(x, \widehat{w}) - F_1(x, w_0)|^2} = \widetilde{O}\left(\sqrt{m/n}\right),$$

where $\mu$ is the input distribution and $w_0$ is the underlying true convolutional filter. Notably, to achieve an $\epsilon$ error, the CNN only needs $\widetilde{O}(m/\epsilon^2)$ samples whereas the FNN needs $\Omega(d/\epsilon^2)$. Since the filter size $m \ll d$, this result clearly justifies the folklore that the convolutional layer is a more compact representation. Furthermore, we complement this upper bound with a minimax lower bound which shows the error bound $\widetilde{O}(\sqrt{m/n})$ is tight up to logarithmic factors.

Next, we consider a one-hidden-layer CNN (see Figure 1b):

$$F_2(x; w, a) = \sum_{\ell=0}^{r-1} a_\ell w^\top \mathsf{P}_s^\ell x, \tag{3}$$

where both the shared convolutional filter $w \in \mathbb{R}^m$ and output weights $a \in \mathbb{R}^r$ are unknown. This architecture is previously considered in Du et al. (2017b). However the focus of that work is to understand the dynamics of gradient descent. Using similar tools as in analyzing a single convolutional filter, we show that the least squares estimator achieves the error bound $\widetilde{O}(\sqrt{(m+r)/n})$ if the ratio between the stride size and the filter size is a constant. Further, we present a minimax lower bound showing that the obtain rate is tight up to logarithmic factors.

To our knowledge, these theoretical results are the first sharp analyses of the statistical efficiency of the CNN. These results suggest that if the input follows a (linear) CNN model, then it can be learned more easily than treating it as a FNN since a CNN model reuses weights.

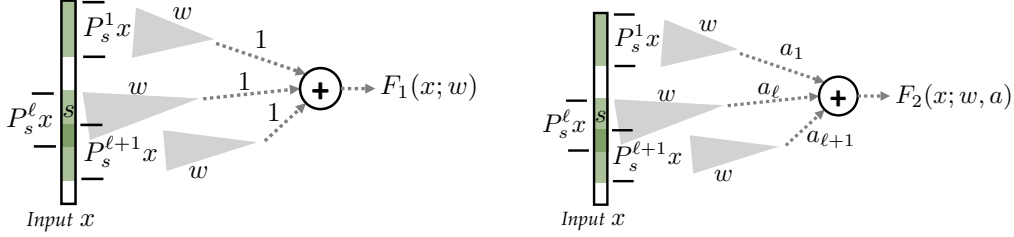

(a) Prediction function formalized in Eq. (1). It consists of a convolutional filter followed by averaged pooling. The convolutional filter is unknown.

(b) Prediction function formalized in Eq. (3) It consists of a convolutional filter followed by a linear prediction layer. Both layers are unknown.

Figure 1: CNN architectures that we consider in this paper.

## 1.1 Comparison with Existing Work

Our work is closely related to the analysis of the generalization ability of neural networks (Arora et al., 2018; Anthony & Bartlett, 2009; Bartlett et al., 2017b,a; Neyshabur et al., 2017; Konstantinos et al., 2017). These generalization bounds are often of the form:

$$L(\theta) - L_{\text{tr}}(\theta) \leqslant D/\sqrt{n} \qquad (4)$$

where $\theta$ represents the parameters of a neural network, $L(\cdot)$ and $L_{\text{tr}}(\cdot)$ represent population and empirical error under some *additive* loss, and $D$ is the model capacity and is finite only if the (spectral) norm of the weight matrix for each layer is bounded. Comparing with generalization bounds based on model capacity, our result has two advantages:

1. If $L(\cdot)$ is taken to be the mean-squared[3] error $\mathbb{E}|\cdot|^2$, Eq. (4) implies an $\widetilde{O}(1/\epsilon^4)$ sample complexity to achieve a standardized mean-square error of $\sqrt{\mathbb{E}|\cdot|^2} \leqslant \epsilon$, which is considerably larger than the $\widetilde{O}(1/\epsilon^2)$ sample complexity we established in this paper.

2. Since the complexity of a model class in regression problems typically depends on the magnitude of model parameters (e.g., $\|w\|_2$), generalization error bounds like Eq. (4) are not scale-independent and deteriorate if $\|w\|_2$ is large. In contrast, our analysis has no dependency on the scale of $w$ and also places no constraints on $\|w\|_2$.

On the other hand, we consider the special case where the neural network model is well-specified and the label is generated according to a neural network with unbiased additive noise (see Eq. (5)) whereas the generalization bounds discussed in this section are typically model agnostic.

## 1.2 Other Related Work

Recently, researchers have been making progress in theoretically understanding various aspects of neural networks, including hardness of learning (Goel et al., 2016; Song et al., 2017; Brutzkus & Globerson, 2017), landscape of the loss function (Kawaguchi, 2016; Choromanska et al., 2015; Hardt & Ma, 2016; Haeffele & Vidal, 2015; Freeman & Bruna, 2016; Safran & Shamir, 2016; Zhou & Feng, 2017; Nguyen & Hein, 2017b,a; Ge et al., 2017b; Zhou & Feng, 2017; Safran & Shamir, 2017; Du & Lee, 2018), dynamics of gradient descent (Tian, 2017; Zhong et al., 2017b; Li & Yuan, 2017), provable learning algorithms (Goel & Klivans, 2017a,b; Zhang et al., 2015), etc.

Focusing on the convolutional neural network, most existing work has analyzed the convergence rate of gradient descent or its variants (Du et al., 2017a,b; Goel et al., 2018; Brutzkus & Globerson, 2017; Zhong et al., 2017a). Our paper differs from them in that we do not consider the computational complexity but only the sample complexity and information theoretical limits of learning a CNN. It is an open question when taking computational budget into account, what is the optimal estimator for CNN.

Convolutional structure has also been studied in the dictionary learning (Singh et al., 2018; Huang & Anandkumar, 2015) and blind de-convolution (Zhang et al., 2017) literature. These papers studied the unsupervised setting where their goal is to recover structured signals from observations generated according to convolution operations whereas our paper focuses on the supervised learning setting with predictor having the convolution structure.

## 1.3 Organization

This paper is organized as follows. In Section 2, we formally setup the problem and assumptions. In Section 3 we present our main theoretical results for learning a convolutional filter (see Eq. (1)). In Section 4 we present our main theoretical results for learning a one-hidden-layer CNN (see Eq. (3)). In Section 5, we use numerical experiments to verify our theoretical findings. We conclude and list future directions in Section 6. Most technical proofs are deferred to the appendix.

## 2 Problem specification and assumptions

Let $\{x_i, y_i\}_{i=1}^n$ be a sample of $n$ training data points, where $x_i \in \mathbb{R}^d$ denotes the $d$-dimensional feature vector of the $i$th data point and $y_i \in \mathbb{R}$ is the corresponding real-valued response. We consider a generic model of

$$y_i = F(x_i; \boldsymbol{w}_0) + \varepsilon_i, \quad \text{where } \mathbb{E}[\varepsilon_i | x_i] = 0. \tag{5}$$

In the model of Eq. (5), $F$ represents a certain network parameterized by a fixed but unknown parameter $\boldsymbol{w}_0$ that takes a $d$-dimensional vector $x_i$ as input and outputs a single real-valued prediction $F(x_i; \boldsymbol{w}_0)$. $\{\varepsilon_i\}_{i=1}^n$ represents stochastic noise inherent in the data, and is assumed to have mean zero. The feature vectors of training data $\{x_i\}_{i=1}^n$ are sampled i.i.d. from an unknown distribution $\mu$ supported on $\mathbb{R}^d$.

Throughout this paper we make the following assumptions:

(A1) *Sub-gaussian noise*: there exists constant $\sigma^2 < \infty$ such that for any $t \in \mathbb{R}$, $\mathbb{E}e^{t\varepsilon_i} \leqslant e^{\sigma^2 t^2/2}$;

(A2) *Sub-gaussian design*: there exists constant $\nu^2 < \infty$ such that for any $a \in \mathbb{R}^d$, $\mathbb{E}_\mu x = 0$ and $\mathbb{E}_\mu \exp\{a^\top x\} \leqslant \exp\{\nu^2 \|a\|_2^2/2\}$;

(A3) *Non-degeneracy*: there exists constant $\kappa > 0$ such that $\lambda_{\min}(\mathbb{E}_\mu xx^\top) \geqslant \kappa$.

We remark that the assumptions (A1) through (A3) are quite mild. In particular, we only impose sub-Gaussianity conditions on the distributions of $x_i$ and $\varepsilon_i$, and do not assume they are generated/sampled from any *exact* distributions. The last non-degeneracy condition (A3) assumes that there is a non-negligible probability mass along any direction of the input distributions. It is very likely to be satisfied after simple pre-processing steps of input data, such as mean removal and whitening of the sample covariance.

We are interested in learning a parameter $\widehat{\boldsymbol{w}}_n$ using a training sample $\{(x_i, y_i)\}_{i=1}^n$ of size $n$ so as to minimize the standardized *population* mean-square prediction error

$$\mathrm{err}_\mu(\widehat{\boldsymbol{w}}_n, \boldsymbol{w}_0; F) = \sqrt{\mathbb{E}_{x \sim \mu} |F(x; \widehat{\boldsymbol{w}}_n) - F(x; \boldsymbol{w}_0)|^2}. \tag{6}$$

## 3 Convolutional filters with average pooling

We first consider a convolutional network with one convolutional layer, one convolutional filter, an average pooling layer and linear activations. More specifically, for a single convolutional filter $w \in \mathbb{R}^m$ of size $m$ and a stride of size $s$, the network can be written as

$$F_1(x; w) = \sum_{\ell=0}^{r-1} w^\top \mathsf{P}_s^\ell x, \tag{7}$$

where $r \approx d/s$ is the total number of times filter $w$ is applied to an input vector $x$, and $\mathsf{P}_s^\ell x := [x_{\ell s+1}, x_{\ell s+2}, \ldots, x_{\ell s+m}]$ is an $m$-dimensional segment of the $d$-dimensional feature vector $x_i$. For simplicity, we assume that $m$ is divisible $s$ and let $J = m/s \in \mathbb{N}$ denote the number of strides within a single filter of size $m$.

## 3.1 The upper bound

Given training sample $\{(x_i, y_i)\}_{i=1}^n$, we consider the following least-squares estimator:

$$\widehat{w}_n \in \arg\min_{w \in \mathbb{R}^m} \frac{1}{n} \sum_{i=1}^n \left(y_i - F_1(x_i; w)\right)^2 . \qquad (8)$$

Note the subscript $n$ which emphasizes that $\widehat{w}_n$ is trained using a sample of $n$ data points. In addition, because the objective is a quadratic function in $w$, Eq. (8) is actually a convex optimization problem and a global optimal solution $\widehat{w}_n$ can be obtained efficiently. More specifically, $\widehat{w}_n$ admits the closed-form solution of $\widehat{w}_n = (\sum_{i=1}^n \widetilde{x}_i \widetilde{x}_i^\top)^{-1} \sum_{i=1}^n y_i \widetilde{x}_i$, where $\widetilde{x}_i = \sum_{\ell=0}^{r-1} \mathsf{P}_s^\ell x_i$ is the stacked version of input feature vector $x_i$.

The following theorem upper bounds the expected population mean-square prediction error $\mathrm{err}_\mu(\widehat{w}_n, w_0; F_1)$ of the least-square estimate $\widehat{w}_n$ in Eq. (8).

**Theorem 1.** *Fix an arbitrary $\delta \in (0, 1/2)$. Suppose (A1) through (A3) hold and $\nu\sqrt{\log(n/\delta)} \geqslant \kappa$, $n \gtrsim \kappa^{-2}\nu^2 m \log(\nu d \log \delta^{-1}) \log(n\delta^{-1})$. Then there exists a universal constant $C > 0$ such that with probability $1 - \delta$ over the random draws of $x_1, \ldots, x_n \sim \mu$,*

$$\mathbb{E}\mathrm{err}_\mu(\widehat{w}_n, w_0; F_1) \leqslant C\sqrt{\frac{\sigma^2 m \log(\kappa^{-1}\nu d \log(\delta^{-1}))}{n}} \qquad \text{conditioned on } x_1, \ldots, x_n. \qquad (9)$$

*Here the expectation is taken with respect to the randomness in $\{\varepsilon_i\}_{i=1}^n$.*

Theorem 1 shows that, with $n = \widetilde{\Omega}(m)$ samples, the expected population mean-square error $\mathrm{err}_\mu(\widehat{w}_n, w_0; F_1)$ scales as $\widetilde{O}(\sqrt{\sigma^2 m/n})$. This matches the $1/\sqrt{n}$ statistical error for classical parametric statistics problems, and also confirms the "parameter count" intuition that the estimation error scales approximately with the number of parameters in a network ($m$ in network $F_1$).

We next briefly explain the strategies we employ to prove Theorem 1. While it's tempting to directly use the closed-form expression $\widehat{w}_n = (\sum_{i=1}^n \widetilde{x}_i \widetilde{x}_i^\top)^{-1} \sum_{i=1}^n y_i \widetilde{x}_i$ to analyze $\widehat{w}_n$, such an approach has two limitations. First, because we consider the *population* mean-square error $\mathrm{err}_\mu(\widehat{w}_n, w_0; F_1)$, such an approach would inevtiably require the analysis of spectral properties (e.g., the least eigenvalue) of $\sum_{i=1}^n \widetilde{x}_i \widetilde{x}_i^\top$, which is very challenging as heavy correlation occurs in $\widetilde{x}_i$ when filters are overlapping (i.e., $s < m$ and $J > 1$). It is likely that strong assumptions such as *exact isotropic* Gaussianity of the feature vectors are needed to analyze the distributional properties $\widetilde{x}_i$ (Qu et al., 2017). Also, such an approach relies on closed-forms of $\widehat{w}_n$ and is difficult to extend to other potential activations such as the ReLU activation. when no closed-form expressions of $\widehat{w}_n$ exist.

To overcome the above difficulties, we adopt a *localized empirical process* approach introduced in (van de Geer, 2000) to upper bound the expected population mean-square prediction error. At the core analysis is an upper bound on the covering number of a *localized parameter set*, with an interesting argument that partitions a $d$-dimensional equivalent regressor for compactification purposes (see Lemmas 2 and 4 in the appendix for details). Our proof does not rely on the exact/closed-form expression of $\widehat{w}_n$, and has the potential to be extended to other activation functions, as we discuss in Section 6. The complete proof of Theorem 1 is placed in the appendix.

## 3.2 The lower bound

We prove the following information-theoretic lower bound on $\mathbb{E}\mathrm{err}_\mu(\widehat{w}_n, w_0)$ of *any* estimator $\widehat{w}_n$ calculated on a training sample of size $n$.

**Theorem 2.** *Suppose $x_1, \ldots, x_n \sim \mathcal{N}(0, I)$ and $\varepsilon_1, \ldots, \varepsilon_n \sim \mathcal{N}(0, \sigma^2)$. Suppose also that $m - s$ is an even number. Then there exists a universal constant $C' > 0$ such that*

$$\inf_{\widehat{w}_n} \sup_{w_0 \in \mathbb{R}^m} \mathbb{E}\mathrm{err}_\mu(\boldsymbol{w}_n, w_0; F_1) \geqslant C'\sqrt{\frac{\sigma^2 m}{n}}. \qquad (10)$$

**Remark 1.** *Theorem 2 is valid for* any *pair of (filter size, stride) combinations $(m, s)$, provided that $m$ is divisible by $s$ and $m - s$ is an even number. The latter requirement is a technical condtion in our proof and is not critical, because one can double the size of $m$ and $s$, and the lower bound in Theorem 2 remains asymptotically on the same order.*

Theorem 2 shows that any estimator $\widehat{w}_n$ computed on a training set of size $n$ must have a worst-case error of at least $\sqrt{\sigma^2 m/n}$. This suggests that our upper error bound in Theorem 1 is tight up to logarithmic factors.

Our proof of Theorem 2 draws on tools from standard information-theoretical lower bounds such as the Fano's inequality (Yu, 1997; Tsybakov, 2009). The high-level idea is to construct a *finite* candidate set of parameters $\mathcal{W} \subseteq \mathbb{R}^m$ and upper bound the Kullback-Leibler (KL) divergence of induced observable distributions and the population prediction mean-square error between parameters in the candidate set $\mathcal{W}$. The complete proof of Theorem 2 is placed in the appendix.

## 4    Convolutional filters with prediction layers

We consider a slightly more complicated convolutional network with two layers: the first layer is a single convolutional filter of size $m$, applied $r$ times to a $d$-dimensional input vector with stride $s$; the second layer is a linear regression prediction layer that produces a single real-valued output.

For such a two-layer network the parameter $\boldsymbol{w}$ can be specified as $\boldsymbol{w} = (w, a)$, where $w \in \mathbb{R}^m$ is the weights in the first-layer convolutional filter and $a \in \mathbb{R}^r$ is the weight in the second linear prediction layer. The network $F_2(x; \boldsymbol{w}) = F_2(x; w, a)$ can then be written as

$$F_2(x; w, a) = \sum_{\ell=0}^{r-1} a_\ell w^\top \mathsf{P}_s^\ell x. \tag{11}$$

Note that in Eq. (11) the vector $a \in \mathbb{R}^r$ is labeled as $a = (a_0, a_1, \ldots, a_{r-1})$ for convenience that matches with the labels of the operator $\mathsf{P}_s^\ell$ for $\ell = 0, \ldots, r-1$.

Compared to network $F_1$ with average pooling, the new network $F_2$ can be viewed as a *weighted* pooling of convolutional filters, with weights $a \in \mathbb{R}^r$ unknown and to be learnt. A graph illustration of the network $F_2$ is given in Figure 1b.

### 4.1    The upper bound

We again consider the least-squares estimator

$$\widehat{\boldsymbol{w}}_n = (\widehat{w}_n, \widehat{a}_n) \in \arg \min_{w \in \mathbb{R}^m, a \in \mathbb{R}^r} \frac{1}{n} \sum_{i=1}^{n} \left(y_i - F_2(x_i; w, a)\right)^2. \tag{12}$$

Again, we use subscript $n$ to emphasize that both $\widehat{w}_n$ and $\widehat{a}_n$ are computed on a training set $\{x_i, y_i\}_{i=1}^n$ of size $n$.

Unlike the least squares problem in Eq. (8) for the $F_1$ network, the optimization problem in Eq. (12) has two optimization variables $w, a$ and is therefore no longer convex. This means that popular optimization algorithms like gradient descent do not necessarily converge to a global minima in Eq. (12). Nevertheless, in this paper we choose to focus on the *statistical* properties of $(\widehat{w}_n, \widehat{a}_n)$ and assume global minimality of Eq. (12) is achieved. On the other hand, because Eq. (12) resembles the matrix sensing problem, it is possible that all local minima are global minima and saddle points can be efficiently escaped (Ge et al., 2017a), which we leave as future work.

The following theorem upper bounds the population mean-square prediction error of any global minimizer $\widehat{\boldsymbol{w}}_n = (\widehat{w}_n, \widehat{a}_n)$ of Eq. (12).

**Theorem 3.** *Fix arbitrary $\delta \in (0, 1/2)$ and define $J := m/s$, where $m$ is the filter size and $s$ is the stride. Suppose (A1) through (A3) hold and $\nu\sqrt{\log(n/\delta)} \geqslant \kappa$, $n \gtrsim \kappa^{-2}\nu^2(rJ + m)\log(\nu d \log \delta^{-1})\log(n\delta^{-1})$. Then there exists a universal constant $C > 0$ such that with probability $1 - \delta$ over the random draws of $x_1, \ldots, x_n \sim \mu$,*

$$\mathbb{E}\mathrm{err}_\mu(\widehat{\boldsymbol{w}}_n, \boldsymbol{w}_0; F_2) \leqslant C\sqrt{\frac{\sigma^2(rJ + m)\log(\kappa^{-1}\nu d \log(\delta^{-1}))}{n}} \qquad \textit{conditioned on } x_1, \ldots, x_n. \tag{13}$$

*Here the expectation is taken with respect to the randomness in $\{\varepsilon_i\}_{i=1}^n$.*

Theorem 3 is proved by a similar localized empirical process arguments as in the proof of Theorem 1. Due to space costraints we defer the complete proof of Theorem 3 to the appendix.

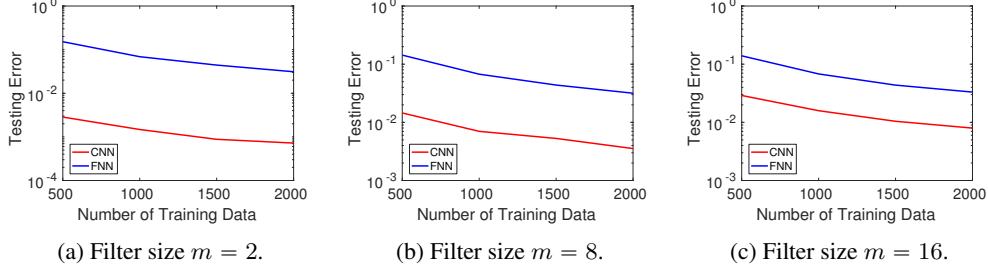

(a) Filter size $m = 2$.  (b) Filter size $m = 8$.  (c) Filter size $m = 16$.

Figure 2: Experiments on the problem of learning a convolutional filter with average pooling described in Section 3 with stride size $s = 1$.

Theorem 3 shows that $\mathrm{err}_\mu(\widehat{\boldsymbol{w}}_n, \boldsymbol{w}_0; F_2)$ can be upper bounded by $\widetilde{O}(\sqrt{\sigma^2(rJ + m)/n})$, provided that at least $n = \widetilde{\Omega}(rJ + m)$ samples are available. Compared to the intuitive "parameter count" of $r + m$ ($r$ parameters for $a$ and $m$ parameters for $w$), our upper bound has an additional multiplicative $J = m/s$ term, which is the number of strides within each $m$-dimensional filter. Therefore, our upper bound only matches parameter counts when $J$ is very small (e.g., non-overlapping filters or fast-moving filters where the stride $s$ is at least a constant fraction of filter size $m$), and becomes large when the stride $s$ is very small, leading to many convolutions being computed.

We conjecture that such an increase in error/sample complexity is due to an inefficiency in one of our key technical lemmas. More specifically, in Lemma 7 in which we derive upper bounds on covering number of localized parameter sets, we use the boundedness and low-dimensionality of each segment of differences of equivalent parameters for compactification purposes; such an argument is not ideal, as it overlooks the correlation between different segments, connected by an $r$-dimensional parameter $a$. A sharper covering number argument would potentially improve the error analysis and achieve sample complexity scaling with $r + m$.

## 4.2  The lower bound

We prove the following information-theoretical lower bound on $\mathbb{E}\mathrm{err}_\mu(\widehat{\boldsymbol{w}}_n, \boldsymbol{w}_0)$ of *any* estimator $\widehat{\boldsymbol{w}}_n = (\widehat{w}_n, \widehat{a}_n)$ calculated on a training sample of size $n$.

**Theorem 4.** *Suppose $x_1, \ldots, x_n \sim \mathcal{N}(0, I)$ and $\varepsilon_1, \ldots, \varepsilon_n \sim \mathcal{N}(0, \sigma^2)$. Then there exists a universal constant $C' > 0$ such that*

$$\inf_{\widehat{\boldsymbol{w}}_n} \sup_{\boldsymbol{w}_0} \mathbb{E}\mathrm{err}_\mu(\widehat{\boldsymbol{w}}_n, \boldsymbol{w}_0; F_2) \geqslant C'\sqrt{\frac{\sigma^2(r + m)}{n}}. \tag{14}$$

Theorem 4 shows that the error of any estimator $\widehat{\boldsymbol{w}}_n$ computed on a training sample of size $n$ must scale as $\sqrt{\sigma^2(r + m)/n}$, matching the parameter counts of $r + m$ for $F_2$. It is proved by reducing the regression problem under $F_2$ to two separate ordinary linear regression problems and invoking classical lower bounds for linear regression models (Wasserman, 2013; Van der Vaart, 1998). A complete proof of Theorem 4 is given in the appendix.

## 5  Experiments

In this section we use simulations to verify our theoretical findings. For all experiments, we let the ambient dimension $d$ be $64$ and the input distribution be Gaussian with mean $0$ and identity covariance. We use the population mean-square prediction error defined in Eq. (6) as the evaluation metric. In all plots, CNN represents using convolutional parameterization corresponding to Eq. (1) or Eq. (3) and FNN represents using fully connected parametrization corresponding to Eq. (2).

In Figure 2 and Figure 3, we consider the problem of learning a convolutional filter with average pooling which we analyzed in Section 3. We vary the number of samples, the dimension of filters and the stride size. Here we compare parameterizing the prediction function as a $d$-dimensional linear predictor and as a convolutional filter followed by average pooling. Experiments show CNN

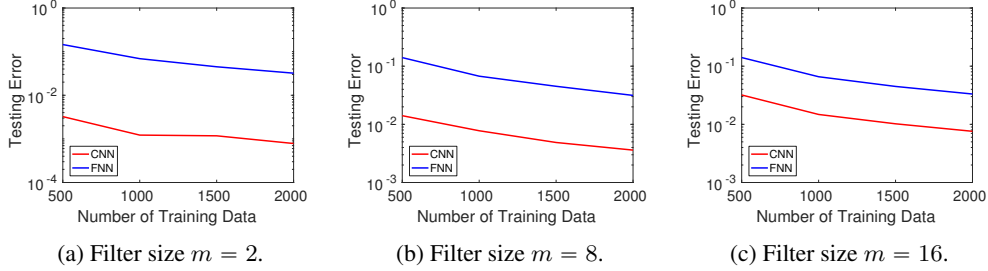

(a) Filter size $m = 2$.  (b) Filter size $m = 8$.  (c) Filter size $m = 16$.

Figure 3: Experiments on the problem of learning a convolutional filter with average pooling described in Section 3 with stride size $s = m$, i.e., non-overlapping.

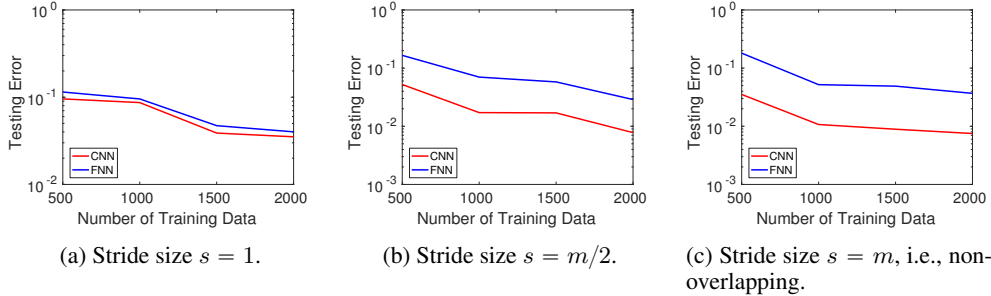

(a) Stride size $s = 1$.  (b) Stride size $s = m/2$.  (c) Stride size $s = m$, i.e., non-overlapping.

Figure 4: Experiment on the problem of one-hidden-layer convolutional neural network with a shared filter and a prediction layer described in Section 4. The filter size $m$ is chosen to be $8$.

parameterization is consistently better than the FNN parameterization. Further, as number of training samples increases, the prediction error goes down and as the dimension of filter increases, the error goes up. These facts qualitatively justify our derived error bound $\widetilde{O}\left(\frac{m}{n}\right)$. Lastly, in Figure 2 we choose stride $s = 1$ and in Figure 3 we choose stride size equals to the filter size $s = m$, i.e., non-overlapping. Our experiment shows the stride does *not* affect the prediction error in this setting which coincides our theoretical bound in which there is no stride size factor.

In Figure 4, we consider the one-hidden-layer CNN model analyzed in Section 4. Here we fix the filter size $m = 8$ and vary the number of training samples and the stride size. When stride $s = 1$, convolutional parameterization has the same order parameters as the linear predictor parameterization ($r = 57$ so $r + m = 65 \approx d = 64$) and Figure 4a shows they have similar performances. In Figure 4b and Figure 4c we choose the stride to be $m/2 = 4$ and $m = 8$ (non-overlapping), respectively. Note these settings have less parameters ($r + m = 23$ for $s = 4$ and $r + m = 16$ for $s = 8$) than the case when $s = 1$ and so CNN gives better performance than FNN.

## 6  Conclusion and Future Directions

In this paper we give rigorous characterizations of the statistical efficiency of CNN with simple architectures. Now we discuss how to extend our work to more complex models and main difficulties.

**Non-linear Activation:**  Our paper only considered CNN with linear activation. A natural question is what is the sample complexity of learning a CNN with non-linear activation like Recitifed Linear Units (ReLU). We find that even without convolution structure, this is a difficult problem. For linear activation function, we can show the empirical loss is a good approximation to the population loss and we used this property to derive our upper bound. However, for ReLU activation, we can find a counter example for any finite $n$, which breaks our Lemma 3. We believe if there is a better understanding of non-smooth activation which can replace our Lemma 3, we can extend our analysis framework to derive sharp sample complexity bounds for CNN with non-linear activation function.

**Multiple Filters:**  For both models we considered in this paper, there is only one shared filter. In commonly used CNN architectures, there are multiple filters in each layer and multiple layers. Note

that if one considers a model of $k$ filters with linear activation with $k > 1$, one can always replace this model by a single convolutional filter that equals to the summation of these $k$ filters. Thus, we can formally study the statistical behavior of wide and deep architectures only after we have understood the non-linear activation function. Nevertheless, we believe our empirical process based analysis is still applicable.

## Acknowledgment

This research was partly funded by AFRL grant FA8750-17-2-0212 and DARPA D17AP00001.

## Footnotes

[2]We use the standard big-O notation in this paper and use $\widetilde{O}(\cdot)$ when we ignore poly-logarithmic factors.

[3]Because the standardized mean-square error $\sqrt{\mathbb{E}|\cdot|^2}$ is not a sum of independent random variables, it is difficult, if not impossible, to apply generalization error bounds directly for $\sqrt{\mathbb{E}|\cdot|^2}$.

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
