[Supplementary Material]

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

## Appendix: Proofs

We first define some notations that will be used throughout the proofs of our results. Note that because the activation function we consider in this paper is the identity mapping, both networks $F_1$ and $F_2$ can be written as a "structured linear regression" model

$$F_{1/2}(x; \boldsymbol{w}) = \langle x, \theta(\boldsymbol{w}) \rangle \quad \text{where} \quad \theta(\boldsymbol{w}) \in \Theta_{1/2} \subseteq \mathbb{R}^d. \tag{15}$$

Here $\Theta_{1/2}$ is a subset of $\mathbb{R}^d$ subject to additional structural constraints corresponding to $F_1$ or $F_2$. We can then define/rewrite "population" and "empirical" mean-square prediction errors as

$$\mathrm{err}_\mu^2(\boldsymbol{w}, \boldsymbol{w}'; F) = \mathbb{E}_\mu \left| \langle x, \theta(\boldsymbol{w}) - \theta(\boldsymbol{w}') \rangle \right|^2 =: \|\theta(\boldsymbol{w}) - \theta(\boldsymbol{w}')\|_\mu^2;$$

$$\widehat{\mathrm{err}}_X^2(\boldsymbol{w}, \boldsymbol{w}'; F) = \frac{1}{n} \sum_{i=1}^n \left| \langle x_i, \theta(\boldsymbol{w}) - \theta(\boldsymbol{w}') \rangle \right|^2 =: \|\theta(\boldsymbol{w}) - \theta(\boldsymbol{w}')\|_X^2;$$

For any set $\Theta \subseteq \mathbb{R}^d$, error parameter $\epsilon > 0$ and a distance metric $d(\cdot, \cdot) : \mathbb{R}^d \times \mathbb{R}^d \to \mathbb{R}^+$, define $N(\epsilon; \Theta, d)$ as the *covering number* of $\Theta$ in $d(\cdot, \cdot)$, which is the size of the smallest finite cover set $H \subseteq \mathbb{R}^d$ such that $\sup_{\phi \in \Theta} \min_{\phi' \in H} d(\phi, \phi') \leq \epsilon$.

## A  Proof of Theorem 1 (upper bound, average pooling)

For the convolutional model with an average pooling layer $F_1(x; w) = \sum_{\ell=0}^{r-1} w^\top \mathsf{P}_s^\ell x$, a linear model $\theta(w) \in \mathbb{R}^d$ as in Eq. (15) can be produced as

$$\theta(w) = \sum_{\ell=0}^{r-1} \mathsf{S}_s^\ell w \quad \text{where} \quad \mathsf{S}_s^\ell w = [\underbrace{0, \ldots, 0}_{\ell s \text{ zeros}}, w_1, \ldots, w_m, 0, \ldots, 0] \in \mathbb{R}^d.$$

Our first lemma is the following "basic inequality", which upper bounds $\widehat{\mathrm{err}}_X^2(\widehat{w}_n, w_0; F) = \|\theta(\widehat{w}_n) - \theta(w_0)\|_X^2$ using a weighted sum of noise variables.

**Lemma 1.** $n \cdot \|\theta(\widehat{w}_n) - \theta(w_0)\|_n^2 \leq 2 \sum_{i=1}^n \varepsilon_i \langle x_i, \theta(\widehat{w}_n) - \theta(w_0) \rangle.$

*Proof.* By definition of $\widehat{w}_n$, we have $\sum_{i=1}^n (y_i - \langle x_i, \theta(\widehat{w}_n) \rangle)^2 \leq \sum_{i=1}^n (y_i - \langle x_i, \theta(w_0) \rangle)^2$. Plugging in $y_i = \langle x_i, \theta(w_0) \rangle + \varepsilon_i$, breaking up the squares and cancelling out the common $\sum_{i=1}^n \varepsilon_i^2$ terms on both sides of the inequality, we have $\sum_{i=1}^n |\langle x_i, \theta(\widehat{w}_n) - \theta(w_0) \rangle|^2 + 2 \sum_{i=1}^n \varepsilon_i \langle x_i, \theta(\widehat{w}_n) - \theta(w_0) \rangle \leq 0$. Re-arranging the terms we proved the lemma. $\square$

We next adopt a localized empirical process approach (van de Geer, 2000) to upper bound $\sum_{i=1}^n \varepsilon_i \langle x_i, \theta(\widehat{w}_n) - \theta(w_0) \rangle$. Define

$$\Theta_{X, F_1} := \{ \theta(w) - \theta(\widetilde{w}) : w, \widetilde{w} \in \mathbb{R}^m, \|\theta(w) - \theta(\widetilde{w})\|_X \leq 1 \}. \tag{16}$$

Re-scaling $(w_0, \widehat{w}_n) \mapsto (w_0, \widehat{w}_n) / \|\theta(w) - \theta(\widetilde{w})\|_X$ and $\varepsilon_i \mapsto \varepsilon_i / \sigma$, we have

$$\frac{1}{n} \sum_{i=1}^n \varepsilon_i \langle x_i, \theta(\widehat{w}_n) - \theta(w_0) \rangle \leq \sigma \|\theta(\widehat{w}_n) - \theta(w_0)\|_X \cdot \sup_{\phi \in \Theta_{X, F_1}} \frac{1}{n} \sum_{i=1}^n \widetilde{\varepsilon}_i \langle x_i, \phi \rangle. \tag{17}$$

For $\phi \in \mathbb{R}^d$ denote $\mathbb{G}_n^X(\phi) := \sum_{i=1}^n \widetilde{\varepsilon}_i \langle x_i, \phi \rangle / \sqrt{n}$ as a random variable with randomness induced by the noise variables $\{\widetilde{\varepsilon}_i\}_{i=1}^n$. Because $\{\widetilde{\varepsilon}_i\}_{i=1}^n$ are i.i.d. centered sub-Gaussian random variables with parameter 1, it is easy to verify that for any $\phi, \phi' \in \mathbb{R}^d$, $\mathbb{G}_n^X(\phi) - \mathbb{G}_n^X(\phi')$ is a centered sub-Gaussian random variable with sub-Gaussian parameter $\gamma^2 \leq \sigma^2 \|\phi - \phi'\|_X^2$. Using Dudley's entropic integral (Dudley, 1967), we have

$$\mathbb{E} \sup_{\phi \in \Theta_{X, F_1}} \mathbb{G}_n^X(\phi) \lesssim \int_0^\infty \sqrt{\log N(\epsilon; \Theta_{X, F_1}, \|\cdot\|_X)} \mathrm{d}\epsilon, \tag{18}$$

Combining Eq. (18) with Lemma 1, we immediately have

$$\mathbb{E}\|\theta(\widehat{w}_n) - \theta(w_0)\|_X \lesssim \sqrt{\frac{1}{n}} \cdot \int_0^\infty \sqrt{\log N(\epsilon; \Theta_{X,F_1}, \|\cdot\|_X)} \mathrm{d}\epsilon. \tag{19}$$

In the rest of the proof we upper bound the integration of covering numbers in Eq. (19). We first consider a "population" version of the localized set $\Theta_{X,F_1}$:

$$\Theta_{\mu,F_1} := \{\theta(w) - \theta(\widetilde{w}) : w, \widetilde{w} \in \mathbb{R}^m, \|\theta(w) - \theta(\widetilde{w})\|_\mu \leq 1\} \tag{20}$$

and upper bounds the covering number $N(\epsilon; \Theta_{\mu,F_1}, \|\cdot\|_X)$. We shall discuss how such an upper bound can be converted into a bound on $N(\epsilon; \Theta_{X,F_1}, \|\cdot\|_X)$ later this section.

We first state two technical lemmas.

**Lemma 2.** *For any $\phi = \theta(w) - \theta(\widetilde{w}) \in \Theta_{\mu,F_1}$ it holds that $\|w - \widetilde{w}\|_2 \leq \kappa^{-1}J^2$, where $J = m/s$.*

*Proof.* By definition, for any $\phi = \theta(w) - \theta(\widetilde{w}) \in \Theta_{\mu,F_1}$ it holds that $\mathbb{E}_\mu|\langle\theta(w) - \theta(\widetilde{w}), x\rangle|^2$. The non-degeneracy condition (A3) of $\mu$ then implies

$$\|\theta(w) - \theta(\widetilde{w})\|_2^2 \leq \kappa^{-2}\mathbb{E}_\mu|\langle\theta(w) - \theta(\widetilde{w}), x\rangle|^2 \leq \kappa^{-2}. \tag{21}$$

Let $\mathrm{Q}_s^0 w, \ldots, \mathrm{Q}_s^{J-1} w$ be $J = m/s$ segments of $w$, each of length $s$. Let also $\mathrm{Q}_s^0 \theta, \ldots, \mathrm{Q}_s^{r-1}\theta$ be $r$ segments of $\theta \in \mathbb{R}^d$, each of length $s$ too. Then

$$\mathrm{Q}_s^\ell[\theta(w) - \theta(\widetilde{w})] = \sum_{\ell'=0}^{\min(J-1,\ell)} \mathrm{Q}_s^\ell(w - \widetilde{w}). \tag{22}$$

Because $\|\theta(w) - \theta(\widetilde{w})\|_2 \leq \kappa^{-1}$, it holds that $\|\mathrm{Q}_s^\ell[\theta(w) - \theta(\widetilde{w})]\|_2 \leq \|\theta(w) - \theta(\widetilde{w})\|_2 \leq \kappa^{-1}$ for all $\ell \in \{0, 1, \ldots, r-1\}$ because $\mathrm{Q}_s^\ell[\theta(w) - \theta(\widetilde{w})]$ partitions $\theta(w) - \theta(\widetilde{w})$ into disjoint segments.

Since $\mathrm{Q}_s^0[\theta(w) - \theta(\widetilde{w})] = \mathrm{Q}_s^0(w - \widetilde{w})$, we know that $\|\mathrm{Q}_s^0(w - \widetilde{w})\|_2 = \|\mathrm{Q}_s^0[\theta(w) - \theta(\widetilde{w})]\|_2 \leq \|\theta(w) - \theta(\widetilde{w})\|_2$. Similarly, because $\mathrm{Q}_s^1[\theta(w) - \theta(\widetilde{w})] = \mathrm{Q}_s^0(w - \widetilde{w}) + \mathrm{Q}_s^1(w - \widetilde{w})$, we have $\|\mathrm{Q}_s^1(w - \widetilde{w})\|_2 \leq \|\mathrm{Q}_s^1[\theta(w) - \theta(\widetilde{w})]\|_2 + \|\mathrm{Q}_s^0(w - \widetilde{w})\|_2 \leq 2\|\theta(w) - \theta(\widetilde{w})\|_2$. Continuing this argument we have $\|\mathrm{Q}_s^\ell(w - \widetilde{w})\|_2 \leq (\ell+1)\|\theta(w) - \theta(\widetilde{w})\|_2$. Subsequently,

$$\|w - \widetilde{w}\|_2 \leq \sum_{\ell=0}^{J-1} \|\mathrm{Q}_s^\ell(w - \widetilde{w})\|_2 \leq J^2 \cdot \|\theta(w) - \theta(\widetilde{w})\|_2 \leq \kappa^{-1}J^2. \tag{23}$$

$\square$

**Lemma 3.** *Fix arbitrary $\delta \in (0, 1/2)$. With probability $1-\delta$ over the random draws of $x_1, \ldots, x_n \sim \mu$, for any $\phi = \theta(w) - \theta(\widetilde{w}), \phi' = \theta(w') - \theta(\widetilde{w}') \in \mathbb{R}^m$ it holds that $\|\phi - \phi'\|_X^2 \lesssim \nu^2\sqrt{d^3\log(1/\delta)} \cdot \|w - \widetilde{w} - w' + \widetilde{w}'\|_2^2$.*

*Proof.* By definition we have that

$$\|\phi - \phi'\|_X^2 = \frac{1}{n}\sum_{i=1}^n |\langle x_i, \phi - \phi'\rangle|^2 \leq \lambda_{\max}(\widehat{\Sigma}_n)\|\phi - \phi'\|_2^2, \tag{24}$$

where $\lambda_{\max}(\widehat{\Sigma}_n)$ is the largest eigenvalue of sample covariance $\widehat{\Sigma}_n = \frac{1}{n}\sum_{i=1}^n x_i x_i^\top$.

Let $\Sigma_0 := \mathbb{E}_\mu xx^\top$ denote the population covariance under $\mu$. Because $\mu$ is sub-Gaussian with parameter $\nu^2$ (A2), by standard concentration inequality of sub-Gaussian sample covariances (e.g., (Vershynin, 2012)) we have with probability $1 - \delta$ that

$$\|\widehat{\Sigma}_n - \Sigma_0\|_{\mathrm{op}} \lesssim \nu^2\sqrt{d\log(1/\delta)/n}. \tag{25}$$

Note that $\|\Sigma_0\|_{\mathrm{op}}$ must also be upper bounded by $\nu^2$ because of the sub-Gaussianity of $\mu$. Therefore, with probability $1 - \delta$

$$\|\phi - \phi'\|_X^2 \lesssim \nu^2\left(1 + \sqrt{\frac{d\log(1/\delta)}{n}}\right)\|\phi - \phi'\|_2^2 \lesssim \nu^2\sqrt{d\log(1/\delta)} \cdot \|\phi - \phi'\|_2^2. \tag{26}$$

Finally, recall the definition that $\phi - \phi' = \theta(w) - \theta(\widetilde{w}) - \theta(w') + \theta(\widetilde{w}') = \sum_{\ell=0}^{r-1} \mathsf{S}_s^\ell (w - \widetilde{w} - w' + \widetilde{w}')$, implying that $\|\phi - \phi'\|_2 \leqslant r \cdot \|w - \widetilde{w} - w' + \widetilde{w}'\|_2 \leqslant d\|w - \widetilde{w} - w' + \widetilde{w}'\|_2$. The lemma is thus proved. $\qquad\square$

We are now ready to state and prove our key covering number lemma of $\Theta_{\mu, F_1}$ in $\|\cdot\|_X$.

**Lemma 4.** *With probability $1 - \delta$ over the random draw of $x_1, \ldots, x_n \sim \mu$, it holds for all $\epsilon > 0$ that $\log N(\epsilon; \Theta_{\mu, F_1}, d_\mu(\cdot, \cdot)) \lesssim m \log(\nu d \log(\delta^{-1})/\epsilon\kappa)$.*

*Proof.* For any $R > 0$ denote $\mathbb{B}_m(R) := \{z \in \mathbb{R}^m : \|z\|_2 \leqslant R\}$ as the centered $m$-dimensional Euclidean ball of radius $R$. By Lemma 2, we know that $\{w - \widetilde{w} : \theta(w) - \theta(\widetilde{w}) \in \Theta_{\mu, F_1}\} \subseteq \mathbb{B}_m(\kappa^{-1}J^2)$.

Let $\mathbf{H} \subseteq \mathbb{R}^m$ be a finite covering set of $\mathbb{B}_m(\kappa^{-1}J^2)$ in $\|\cdot\|_2$ up to a difference precision parameter $\epsilon' > 0$ to be specified later, meaning that $\sup_{\Delta w \in \mathbb{B}_m(\kappa^{-1}J^2)} \min_{\Delta w' \in \mathbf{H}} \|\Delta w - \Delta w'\|_2 \leqslant \epsilon'$. Again using the standard covering number of $m$-dimensional unit balls (e.g., (van de Geer, 2000)), the size of $|\mathbf{H}|$ can be upper bounded by $\log |\mathbf{H}| \lesssim m \log(J^2/\kappa\epsilon')$.

Let $\Phi(\mathbf{H}) := \{\phi' = \theta(w') - \theta(\widetilde{w}') : w' - \widetilde{w}' \in \mathbf{H}\}$ be the induced $d$-dimensional parameter sets by $\mathbf{H}$. Clearly $\log |\Phi(\mathbf{H})| \leqslant \log |\mathbf{H}|$. On the other hand, by Lemma 3 and the fact that $\{w - \widetilde{w} : \theta(w) - \theta(\widetilde{w}) \in \Theta_{\mu, F_1}\} \subseteq \mathbb{B}_m(\kappa^{-1}J^2)$, we have $\sup_{\phi \in \Theta_{\mu, F_1}} \min_{\phi' \in \Phi(\mathbf{H})} \|\phi - \phi'\|_X \lesssim \nu(d^3 \log(1/\delta))^{1/4} \cdot \epsilon'$. Putting $\epsilon' \asymp \epsilon/\nu(d^3 \log(1/\delta))^{1/4}$ we proved the lemma. $\qquad\square$

Finally, we show how an upper bound on $N(\epsilon; \Theta_{\mu, F_1}, \|\cdot\|_X)$ can be turned into an upper bound on $N(\epsilon'; \Theta_{X, F_1}, \|\cdot\|_X)$ for a potentially different precision parameter $\epsilon'$. This is done by considering the following "restricted eigenvalue" (Bickel et al., 2009) type conditions.

**Lemma 5.** *Fix arbitrary $\delta \in (0, 1/2)$. If $\nu\sqrt{\log(n/\delta)} \geqslant \kappa$ and $n \gtrsim \kappa^{-2}\nu^2 m \log(\kappa^{-1}\nu d \log \delta^{-1}) \log(n\delta^{-1})$, then with probability $1 - \delta$ we have that $\|\phi\|_X^2 \geqslant 1/2\|\phi\|_\mu^2$ uniformly for all $\phi \in \Theta_{\mu, F_1}$.*

*Proof.* Because both $\|\cdot\|_\mu$ and $\|\cdot\|_X$ are linear (i.e., $\|a\phi\| = a\|\phi\|$ for all $a \in \mathbb{R}$), it suffices to consider $\phi \in \Theta_{\mu, F_1}$ with $\|\phi\|_\mu = 1$ only.

We first consider the case of *fixed* $\phi \in \Theta_{\mu, F_1}$. Because $\|\phi\|_\mu^2 = 1$, we have $\|\phi\|_2^2 \leqslant \kappa^{-2}$ thanks to (A3). In addition, because $x_1, \ldots, x_n \sim \mu$ are independent sub-Gaussian random vectors with sub-Gaussian parameter $\nu^2$, we have with probability $1 - 0.1\delta$ that $\max_i \|x_i\|_2 \lesssim \nu\sqrt{\log(n/\delta)}$. Subsequently, $\max_i |\langle \phi, x_i \rangle|^2 \lesssim \kappa^{-1}\nu\sqrt{\log(n/\delta)}$. Conditioned on this event, using Hoeffding's concentration inequality (Hoeffding, 1963) we have with probabilty $1 - \delta'$ that

$$\left| \|\phi\|_X^2 - \|\phi\|_\mu^2 \right| = \left| \frac{1}{n} \sum_{i=1}^n |\langle x_i, \phi \rangle|^2 - \mathbb{E}_\mu |\langle x, \phi \rangle|^2 \right| \lesssim \kappa^{-1}\nu\sqrt{\log(n/\delta)} \cdot \sqrt{\frac{\log(1/\delta')}{n}}. \qquad (27)$$

Next consider a finite covering set $\mathbf{H}(\epsilon')$ of $\Theta_{\mu, F_1}$ in distance metric $\|\cdot\|_X$ up to a precision parameter $\epsilon' > 0$ to be specified later; that is, $\sup_{\phi \in \Theta_{\mu, F_1}} \min_{\phi' \in \mathbf{H}(\epsilon')} \|\phi - \phi'\|_X \leqslant \epsilon'$. Lemma 4 guarantees the existence of such a covering set with size $\log |\mathbf{H}(\epsilon')| \lesssim m \log(\nu d \log(\delta^{-1})/\epsilon'\kappa)$ with probability $1 - 0.1\delta$. On the other hand,

$$\left| \|\phi\|_X^2 - \|\phi'\|_X^2 \right| = \left| \frac{1}{n} \sum_{i=1}^n \langle x_i, \phi \rangle^2 - \langle x_i, \phi' \rangle^2 \right| = \left| \frac{1}{n} \sum_{i=1}^n \langle x_i, \phi + \phi' \rangle \langle x_i, \phi - \phi' \rangle \right|$$

$$\leqslant \sqrt{\frac{1}{n} \sum_{i=1}^n |\langle x_i, \phi + \phi' \rangle|^2} \sqrt{\frac{1}{n} \sum_{i=1}^n |\langle x_i, \phi - \phi' \rangle|^2}$$

$$\leqslant \|\phi + \phi'\|_X \cdot \|\phi - \phi'\|_X \leqslant (2\|\phi\|_X + \|\phi - \phi'\|_X) \cdot \|\phi - \phi'\|_X.$$

Because $\|\phi\|_\mu = 1$, we have $\|\phi\|_X \leqslant \max_i \|x_i\|_2 \cdot \|\phi\|_2 \lesssim \kappa^{-1}\nu\sqrt{\log(n/\delta)}$ with probability $1 - 0.1\delta$ uniformly for all $\phi \in \Theta_{\mu, F_1}$. Subsequently,

$$\left| \|\phi\|_X^2 - \|\phi'\|_X^2 \right| \lesssim (\kappa^{-1}\nu\sqrt{\log(n/\delta)} + \epsilon')\epsilon'. \qquad (28)$$

Setting $\epsilon' \asymp O(1)$, $\delta' = 0.1\delta/|\mathbf{H}(\epsilon')|$ and combining Eqs. (27,28) we proved the desired lemma. $\square$

**Corollary 1.** *Under the same conditions in Lemma 4, $N(\epsilon; \Theta_{X,F_1}, \|\cdot\|_X) \leqslant N(\epsilon/2; \Theta_{\mu,F_1}, \|\cdot\|_X)$.*

Combining Lemmas 1, 4 and Corollary 1 we have with probability $1 - \delta$ that

$$\mathbb{E}\|\theta(\widehat{w}_n) - \theta(w_0)\|_X \lesssim \sqrt{\frac{\sigma^2 m \log(\kappa^{-1}\nu d \log(\delta^{-1}))}{n}}. \tag{29}$$

Invoking Lemma 5 again we can upper bound $\|\theta(\widehat{w}_n) - \theta(w_0)\|_\mu = \mathrm{err}_\mu(\widehat{w}_n, w_0; F)$.

## B   Proof of Theorem 2 (lower bound, average pooling)

We use the standard *Fano's inequality* (Yu, 1997; Tsybakov, 2009) to prove the minimax lower bound in Theorem 2. Below we state a commonly used variant of Fano's inequality from (Tsybakov, 2009), also known as the *Tsybakov's master theorem*:

**Lemma 6.** *Let $\mathcal{W} = (w_0, w_1, \ldots, w_M)$ be a finite collection of parameters and let $P_j$ be the distribution induced by parameter $w_j$, for $j \in \{0, \ldots, M\}$. Let also $D : \mathcal{W} \times \mathcal{W} \to \mathbb{R}^+$ be a semi-distance. Suppose the following conditions hold:*

1. $D(w_j, w_k) \geqslant 2\rho > 0$ *for all $j, k \in \{0, \ldots, M\}$;*

2. $P_j \ll P_0$ *for every $j \in \{1, \ldots, M\}$;* [4]

3. $\frac{1}{M} \sum_{j=1}^M \mathrm{KL}(P_j \| P_0) \leqslant \gamma \log M$;

*then the following bound holds:*

$$\inf_{\widehat{w}} \sup_{w_j \in \mathcal{W}} \Pr_j \left[ D(\widehat{w}, w_j) \geqslant \rho \right] \geqslant \frac{\sqrt{M}}{1 + \sqrt{M}} \left( 1 - 2\gamma - 2\sqrt{\frac{\gamma}{\log M}} \right). \tag{30}$$

Recall that $J = m/s$ is the number of strides in each filter, which is assumed to be an integer. We consider two cases separately.

***The non-overlapping case*** $J = 1$.   Construct subset $\mathcal{Z} = \{z_1, \ldots, z_m\} \in \{-1, 1\}^m$ such that

1. For every $z \in \mathcal{Z}$, $\sum_{i=1}^n z_i = 0$;
2. For every distinct pairs of $z, z' \in \mathcal{Z}$, $\Delta_H(z, z') = \sum_{i=1}^m \mathbf{1}\{z_i \neq z_i'\} \geqslant d/16$.

Using classical constructions of separable constant-weight codes (e.g., (Wang & Singh, 2016, Lemma 9), (Graham & Sloane, 1980, Theorem 7)), such a subset $\mathcal{Z}$ exists with size $\log|\mathcal{Z}| \gtrsim m$.

Construct $\mathcal{W} = \{w_0, w_1, \ldots, w_M\} \subseteq \mathbb{R}^m$ as $w_0 = 0$ and $w_j = \delta z_j$ for $j \in \{1, \ldots, M\}$ and some $\delta > 0$ to be specified later. Recall that $\theta(w) = \sum_{\ell=0}^{r-1} \mathsf{S}_s^\ell w$. Note also that $x_1, \ldots, x_n \sim \mathcal{N}(0, I)$ and $\varepsilon_1, \ldots, \varepsilon_n \sim N(0, 1)$. The conditions in Lemma 6 can be verified below:

1. For every $w, w' \in \mathcal{W}$, $D(w, w') := \|\theta(w) - \theta(w')\|_2 \geqslant \sqrt{r}\delta \cdot d/12 \gtrsim \delta\sqrt{r}d$;
2. For every $w_j \in \{w_1, \ldots, w_M\}$, we have $P_j \ll P_0$ and furthermore $\mathrm{KL}(P_j \| P_0) = \mathbb{E}\sum_{i=1}^n |\langle x_i, \theta(w_j) - \theta(w_0)\rangle|^2/2\sigma^2 = n\|\theta(w_j) - \theta(w_0)\|_2^2/\sigma^2 \lesssim nr\delta^2/\sigma^2$.

Setting $\delta \asymp \sqrt{\sigma^2 m/nr}$ and invoking Lemma 6 we have

$$\inf_{\widehat{w}} \sup_{w_j \in \mathcal{W}} \Pr_j \left[ \|\widehat{w} - w_j\|_2 \geqslant c_0 \sqrt{\frac{\sigma^2 m}{n}} \right] \geqslant \frac{1}{4}. \tag{31}$$

Theorem 2 is then proved by applying the Markov's inequality and noting that $\mathrm{err}_\mu(\widehat{w}, w) = \sqrt{\mathbb{E}_\mu |\langle x, \widehat{w} - w\rangle|^2} = \|\widehat{w} - w\|_2$.

***The overlapping case $J > 1$.*** Construct subset $\mathcal{Z} = \{z_1, \ldots, z_m\} \in \{-1, 1\}^m$ such that

1. For every $z \in \mathcal{Z}$, $\sum_{i=1}^n z_i = 0$ and $Q_s^{J-1} z = 0$;
2. For every distinct pairs of $z, z' \in \mathcal{Z}$, $\Delta_H(z, z') = \sum_{i=1}^m \mathbf{1}\{z_i \neq z_i'\} \geqslant d/16$.

Using again the construction of separable constant-weight codes, such a subset $\mathcal{Z}$ exists with size $\log |\mathcal{Z}| \gtrsim m - s \geqslant m/2$.

For every $z \in \mathcal{Z} \subseteq \mathbb{R}^m$, construct $w(z) \in \mathbb{R}^m$ as follows:

$$Q_s^\ell w := Q_s^\ell z - \sum_{\ell'=0}^{\ell-1} Q_s^{\ell'} z \qquad \ell = 0, 1, \ldots, J-1, \tag{32}$$

It is then easy to verify that $\theta(w(z)) = \sum_{\ell=0}^{r-1} S_s^\ell w(z) = (z_1, \ldots, z_m, 0, \ldots, 0)$.

Construct $\mathcal{W} = \{w_0, w_1, \ldots, w_M\} \subseteq \mathbb{R}^m$ as $w_0 = 0$ and $w_j = \delta' z_j$ for $j \in \{1, \ldots, M\}$ and some $\delta' > 0$ to be specified later. The analysis in the non-overlapping case remains valid:

1. For every $w, w' \in \mathcal{W}$, $D(w, w') := \|\theta(w) - \theta(w')\|_2 =\geqslant \delta' \cdot d/12 \gtrsim \delta\sqrt{r}d$;
2. For every $w_j \in \{w_1, \ldots, w_M\}$, we have $P_j \ll P_0$ and furthermore $\mathrm{KL}(P_j \| P_0) = \mathbb{E} \sum_{i=1}^n |\langle x_i, \theta(w_j) - \theta(w_0)\rangle|^2/2\sigma^2 = n\|\theta(w_j) - \theta(w_0)\|_2^2/\sigma^2 \lesssim n\delta^2/\sigma^2$.

Setting $\delta' \asymp \sqrt{\sigma^2 m/n}$ we complete the proof.

## C  Proof of Theorem 3 (upper bound, prediction layers)

We use a similar framework as in the proof of Theorem 1 to prove Theorem 3. For the convolutional network with prediction layers, the parameterization $\theta(w, a) \in \mathbb{R}^d$ takes the form of

$$\theta(w, a) = \sum_{\ell=0}^{r-1} a_\ell S_s^\ell w. \tag{33}$$

Deifne

$$\Theta_{X, F_2} := \{\theta(w, a) - \theta(\widetilde{w}, \widetilde{a}) : w, \widetilde{w} \in \mathbb{R}^m, a, \widetilde{a} \in \mathbb{R}^r, \|\theta(w, a) - \theta(\widetilde{w}, \widetilde{a})\|_X \leqslant 1\}. \tag{34}$$

Using the same basic inequality and Dudley's entropic integral as in the proof of Theorem 1, we have

$$\mathbb{E}\|\theta(\widehat{w}_n, \widehat{a}_n) - \theta(w_0, a_0)\|_X \lesssim \sqrt{\frac{1}{n}} \cdot \int_0^\infty \sqrt{\log N(\epsilon; \Theta_{X, F_2}, \|\cdot\|_X)} d\epsilon. \tag{35}$$

We similarly also consider a population version of $\Theta_{X, F_1}$:

$$\Theta_{\mu, F_2} := \{\theta(w, a) - \theta(\widetilde{w}, \widetilde{a}) : w, \widetilde{w} \in \mathbb{R}^m, a, \widetilde{a} \in \mathbb{R}^r, \|\theta(w, a) - \theta(\widetilde{w}, \widetilde{a})\|_\mu \leqslant 1\}. \tag{36}$$

The following lemma upper bounds the covering number of $\Theta_{\mu, F_2}$ with respect to $\|\cdot\|_X$.

**Lemma 7.** *With probability $1 - \delta$ over the random draw of $x_1, \ldots, x_n \sim \mu$, it holds for all $\epsilon > 0$ that $\log N(\epsilon; \Theta_{\mu, F_2}, \|\cdot\|_X) \lesssim (rJ + m) \log(\kappa^{-1}\nu d \log \delta^{-1})$.*

*Proof.* Consider any $\phi = \theta(w, a) - \theta(\widetilde{w}, \widetilde{a}) \in \Theta_{\mu, F_2}$. By definition, we know that $\|\phi\|_\mu \leqslant 1$, and therefore $\|\phi\|_2 \leqslant \kappa^{-1}$ thanks to the non-degeneracy condition (A3).

Let $Q_s^0 \phi, \ldots, Q_s^{r-1} \phi$ be the $r$ disjoint segments of $\phi \in \mathbb{R}^d$, each of length $s$. Let also $Q_s^0 w, \ldots, Q_s^{J-1} w$ be the $J$ disjoint segments of $w \in \mathbb{R}^m$ each of length $s$. Denote for convenience that $a_t = a_{t \bmod r}$ and $\widetilde{a}_t = \widetilde{a}_{t \bmod r}$, which extends the subscripts of $a$ and $\widetilde{a}$ to $\mathbb{Z}$. Then

$$Q_s^\ell \phi = \sum_{j=0}^{J-1} a_{\ell+j} Q_s^j w - \widetilde{a}_{\ell+j} Q_s^j \widetilde{w}, \qquad \ell = 0, 1, \ldots, r-1. \tag{37}$$

Subsequently, for every $\ell$ we have $\mathsf{Q}_s^\ell \phi \in \mathrm{span}\{\mathsf{Q}_s^j w, \mathsf{Q}_s^j \widetilde{w}\}_{j=0}^{J-1}$, a linear subspace in $\mathbb{R}^s$ of dimension at most $2J$. Note also that $\|\mathsf{Q}_s^\ell \phi\|_2 \leqslant \|\phi\|_2 \leqslant \kappa^{-1}$. We then have (recall that $\mathbb{B}_s(R)$ denotes the centered $m$-dimensional Euclidean ball of radius $R$)

$$\Theta_{\mu,F_2} \subseteq \{\phi \in \mathbb{R}^m : \exists \mathcal{S} \subseteq \mathbb{R}^s, \dim(S) \leqslant J \ s.t. \ \mathsf{Q}_s^\ell \phi \in \mathcal{S} \cap \mathbb{B}_s(\kappa^{-1})\} =: \widetilde{\Theta}. \tag{38}$$

Our construction of covering sets of $\widetilde{\Theta}$ (and therefore also $\Theta_{\mu,F_2}$) can be divided into two steps. As a first step, we construct covering set $\mathbf{S} = \{\mathcal{S}_1^*, \ldots, \mathcal{S}_N^*\}$ such that each $\mathcal{S}_k^*, \ell \in [N]$ is linear subspace in $\mathbb{R}^s$ of dimension at most $2J$, and furthermore for any linear subspace $\mathcal{S} \subseteq \mathbb{R}^s, \dim(\mathcal{S}) \leqslant 2J$,

$$\min_{\mathcal{S}_k^* \in \mathbf{S}} \sup_{z \in \mathcal{S} \cap \mathbb{B}_s(1)} \inf_{z' \in \mathcal{S}_k^* \cap \mathbb{B}_s(1)} \|z - z'\|_2 \leqslant \epsilon', \tag{39}$$

where $\epsilon' > 0$ is an error tolerance parameter to be specified later. The following proposition gives an upper bound on the size of such coverings, which is proved later.

**Proposition 1.** *There exists $\mathbf{S}$ satisfying Eq. (39); furthermore, $\log |\mathbf{S}| \lesssim m \log(d/\epsilon')$.*

The next step is to construct, for each $\mathcal{S}_k^* \in \mathbf{S}$, a covering $\mathbf{H}(\mathcal{S}_k^*) = \{u_1^*, \ldots, u_T^*\} \subseteq \mathcal{S}_k^* \cap \mathbb{B}_s(1)$ satisfying

$$\sup_{u \in \mathcal{S}_k^* \cap \mathbb{B}_s(1)} \min_{u_t^* \in \mathbf{H}(\mathcal{S}_k^*)} \|u - u_t^*\|_2 \leqslant \epsilon'', \tag{40}$$

where $\epsilon'' > 0$ is another error tolerance parameter to be specified later. The following proposition gives an upper bound on the size of such coverings. Its proof is also given later.

**Proposition 2.** *There exists $\mathbf{H}(\mathcal{S}_k^*)$ satisfying Eq. (40); furthermore, $\log |\mathbf{H}(\mathcal{S}_k^*)| \lesssim J \log(1/\epsilon'')$.*

We now construct our final covering set as follows:

$$\mathbf{C} := \bigcup_{\mathcal{S}_k^* \in \mathbf{S}} \{\phi \in \mathbb{R}^d : \mathsf{Q}_s^0 \phi, \ldots, \mathsf{Q}_s^{r-1} \phi \in \mathbf{H}(\mathcal{S}_k^*)\}. \tag{41}$$

For any $\phi \in \widetilde{\Theta}$ corresponding to linear subspace $\mathcal{S}$ in $\mathbb{R}^s, \dim(\mathcal{S}) \leqslant 2J$, one first finds $\mathcal{S}_k^* \in \mathbf{S}$ that best approximates $\mathcal{S}$ in the sense of Eq. (39). Then for each $\mathsf{Q}_s^\ell \phi, \ell \in \{0, \ldots, r-1\}$, one can find $u_\ell^* \in \mathbf{H}(\mathcal{S}_k^*)$ that minimizes $\|\mathsf{Q}_s^\ell \widetilde{\phi} - \widetilde{u}_\ell^*\|_2$ where $\widetilde{u}_\ell^* = \|\mathsf{Q}_s^\ell \widetilde{\phi}\|_2 \widetilde{u}_\ell^*$ and $\mathsf{Q}_s^\ell \widetilde{\phi} \in \mathcal{S}_\ell^*$ minimizes $\|\mathsf{Q}_s^\ell (\phi - \widetilde{\phi})\|_2$. Define $\phi^* \in \mathbb{R}^d$ as $\mathsf{Q}_s^\ell \phi^* = \widetilde{u}_\ell^*$. Then

$$\|\phi - \phi^*\|_2 \leqslant \sum_{\ell=0}^{r-1} \|\mathsf{Q}_s^\ell (\phi - \widetilde{\phi})\|_2 + \|\mathsf{Q}_s^\ell \widetilde{\phi} - \widetilde{u}_\ell^*\|_2 \leqslant r\kappa^{-1}(\epsilon' + \epsilon''), \tag{42}$$

where the last inequality holds by a scaling argument and the fact that $\|\mathsf{Q}_s^\ell \phi\|_2 \leqslant \kappa^{-1}$. Finally, because $x_1, \ldots, x_n \sim \mu$ are independent sub-Gaussian random vectors, using Eq. (25) we have with probability $1 - \delta$ that

$$\|\phi - \phi^*\|_X \lesssim \nu^2 \sqrt{d \log(1/\delta)} \cdot \|\phi - \phi^*\|_2 \leqslant \kappa^{-1}\nu^2 \sqrt{d^3 \log(1/\delta)} \cdot (\epsilon' + \epsilon''). \tag{43}$$

Setting $\epsilon' = \epsilon'' \asymp \epsilon/(\kappa^{-1}\nu^2\sqrt{d^3 \log(1/\delta)})$ we proved that $\mathbf{C}$ is a valid covering set of $\widetilde{\Theta}$ (and also $\Theta_{\mu,F_2}$) in $\|\cdot\|_X$ up to precision $\epsilon$.

Finally we count the number of elements in $\mathbf{C}$. By Propositions 1 and 2, we have $\log |\mathbf{S}| \lesssim J \log(\kappa^{-1}\nu d \log \delta^{-1})$ and $\log |\mathbf{H}(\mathcal{S}_\ell^*)| \lesssim J \log(\kappa^{-1}\nu d \log \delta^{-1})$. By construction of $\mathbf{C}$, we have $|\mathbf{C}| \leqslant |\mathbf{S}| \times \max_{\mathcal{S}_k^* \in \mathbf{S}} |\mathbf{H}(\mathcal{S}_k^*)|^r$. Subsequently, $\log |\mathbf{C}| \lesssim (rJ + m) \log(\kappa^{-1}\nu d \log \delta^{-1})$. $\qquad\square$

*Proof of Proposition 1.* If $2J \geqslant s$ then the proposition clearly holds. So we shall only prove the proposition in cases where $2J \leqslant s$.

Let $\mathcal{U}, \mathcal{V}$ be two linear subspace of $\mathbb{R}^s$ of dimension at most $2J$. Let $U, V \in \mathbb{R}^{s \times 2J}$ be the corresponding orthonormal basis of $\mathcal{U}$ and $\mathcal{V}$, with orthogonal columns. Any $u \in \mathcal{U} \cap \mathbb{B}_s(1)$ can then be written as $u = U\alpha$ with $\|\alpha\|_2 = 1$. Consider $v := V\alpha$. It is easy to verify that $v \in \mathcal{V} \cap \mathbb{B}_s(1)$. In addition, $\|u - v\|_2 = \|(U - V)\alpha\|_2 \leqslant \|U - V\|_{\mathrm{op}} \leqslant \|U - V\|_F$. Subsequently, a covering of $\{U \in \mathbb{R}^{2J \times s} : \|U\|_F \leqslant \sqrt{2J}\|U\|_{\mathrm{op}} = \sqrt{2J}\}$ in $\|\cdot\|_F$ up to precision $\epsilon'$ implies a covering in the sense of Proposition 1. By viewing $U$ as a $(2J \times s)$-dimensional vector in the Euclidean space, it is easy to see that such a cover exists with size $\log N \lesssim (2Js) \log(2J/\epsilon') \lesssim Js \log(d/\epsilon') = m \log(d/\epsilon')$. $\qquad\square$

*Proof of Proposition 2.* If $2J \geqslant s$ then the proposition clearly holds, because one only needs to invoke classical coverings of $\mathbb{B}_s(1)$. So we shall only consider cases where $2J \leqslant s$.

Let $U \in \mathbb{R}^{2J \times s}$ be an orthonormal basis of $\mathcal{S}_k^*$. Any $u \in \mathcal{S}_k^* \cap \mathbb{B}_s(1)$ can be written as $u = U\alpha$ for some $\|\alpha\|_2 \leqslant 1$. For any other $\beta \in \mathbb{R}^{2J}$, $\|\beta\|_2 \leqslant 1$, consider $v = U\beta$. It is easy to verify that $v \in \mathcal{S}_k^* \cap \mathbb{B}_s(1)$. Furthermore, $\|u - v\|_2 = \|U(\alpha - \beta)\|_2 \leqslant \|\alpha - \beta\|_2$. Therefore, a covering of $\mathbb{B}_{2J}(1)$ in $\|\cdot\|_2$ up to precision $\epsilon''$ implies a covering in the sense of Proposition 2. On the other hand, standard covering number arguments (e.g., (van de Geer, 2000)) show that such a covering exists with size $\log N \lesssim 2J \log(1/\epsilon'')$. $\square$

Finally, we show that an upper bound on $\log N(\epsilon; \Theta_{\mu,F_1}, \|\cdot\|_X)$ implies an upper bound on $\log N(\epsilon'; \Theta_{X,F_1}, \|\cdot\|_X)$ for a potentially different precision parameter $\epsilon'$. Similar to the proof of Theorem 1, the following lemma establishes a restricted eigenvalue type condition for network $F_2$ and the parameter spaces $\Theta_{\mu,F_1}, \Theta_{X,F_1}$ it induces.

**Lemma 8.** *Fix arbitrary* $\delta \in (0, 1/2)$. *If* $\nu\sqrt{\log(n/\delta)} \geqslant \kappa$ *and* $n \gtrsim \kappa^{-2}\nu^2(rJ + m)\log(\kappa^{-1}\nu d \log \delta^{-1})\log(n\delta^{-1})$, *then with probability* $1 - \delta$ *we have that* $\|\phi\|_X^2 \geqslant 1/2\|\phi\|_\mu^2$ *uniformly for all* $\phi \in \Theta_{\mu,F_2}$.

The proof of Lemma 8 is identical to the proof of Lemma 5 except that a different covering number lemma is invoked; therefore we omit the proof. Combining Lemmas 7, 8 with Eq. (35) we proved Theorem 3.

## D   Proof of Theorem 4 (lower bound, prediction layers)

Because $r + m \leqslant 2\max(r, m)$, it suffices to prove minimax lower bounds of $\sqrt{\sigma^2 m/n}$ and $\sqrt{\sigma^2 r/n}$ separately.

First consider $a_0 = (1, 0, \ldots, 0)$ and $w_0 \in \mathbb{R}^m$ free to vary. Then $F_2(x; a_0, w_0) = w_0^\top \mathsf{P}_s^0 x$ reduces to a standard linear regression problem with $m$ covariates. It is a classical result (e.g., (Van der Vaart, 1998)) that the minimax mean-square error of learning an $m$-dimensional linear predictor is $m/n$; more specifically,

$$\inf_{\widehat{w}_n} \sup_{w_0 \in \mathbb{R}^m} \mathbb{E}\|\widehat{w}_n - w\|_2 \gtrsim \sqrt{\sigma^2 m/n}. \tag{44}$$

On the other hand, because $a_0 = (1, 0, \ldots, 0)$ and $x_1, \ldots, x_n \sim \mathcal{N}(0, I)$, we have that $\mathrm{err}_\mu^2(\widehat{\boldsymbol{w}}_n, \boldsymbol{w}_0; F_2) = \|\widehat{w}_n - w_0\|_2^2$. Therefore, Eq. (44) implies a $\sqrt{\sigma^2 m/n}$ lower bound on the minimax mean-square error $\mathbb{E}\mathrm{err}_\mu(\widehat{\boldsymbol{w}}_n, \boldsymbol{w}_0; F_2)$.

Next consider $w_0 = (1, 0, \ldots, 0)$ and $a \in \mathbb{R}^r$ free to vary. Denote $\widetilde{x} := (x_0, x_s, \ldots, x_{(r-1)s}) \in \mathbb{R}^r$. Then $F_2(x; a_0, w_0) = a_0^\top \widetilde{x}$. Also note that $\widetilde{x} \sim \mathcal{N}(0, I_r)$ because $x \in \mathcal{N}(0, I_d)$. Using the same analysis above we can establish a $\sqrt{\sigma^2 r/n}$ lower bound on the minimax mean-square error $\mathbb{E}\mathrm{err}_\mu(\widehat{\boldsymbol{w}}_n, \boldsymbol{w}_0; F_2)$.

Combining both cases we complete the proof of Theorem 4.