[Reviews · NeurIPS 2018]

Reviewer 1



The authors consider the number of samples needed to achieve an error epsilon in the context of learning an m-dimensional convolutional filter as well as one followed by a linear projection. This is motivated by a desire to rigorously understand the empirical success of CNNs. This paper seems technically correct, yet I believe the setting is very far from real CNNs to the point where it’s not clear if the results will be impactful. The authors only consider a linear convolution layer, which corresponds to a wiener filtering-like operation according to their model, for removing noise for estimating the label. My concern is the motivation, the novelty and the assumptions. For the assumptions, I am not sure that the non degeneracy assumption (A3) is that easy to constraint. Just whitening looks complex, for example in the case of image: either one could use a basis a priori such as a wavelet basis but then this operation is quite complex wrt the tools used in this paper; a second option would be to whiten directly the signal that might be approximative somehow for very large signals For novelty, the authors relate their work to relevant and very recent work on understanding and characterizing deep learning, however I am surprised there is no discussions of any classic signal processing/adaptive filters, the linear restriction in the model and the squared loss gives me a sense that there is a literature from signal processing not considered. In general the lack of literature before 2015 seems odd. I'd like the authors to relate their work to classic signal processing which considers in detail learning linear filters. Additionally I would suggest a relationship to the classic results such as (Barron, 1994) on one-hidden-layer networks. Regarding the motivation, while I agree the results is interesting, since as the authors admit the result is quite distant from the non-linear activation case and it is unclear to me if this result will be useful.

Reviewer 2



This paper characterizes the sample complexity of learning a CNN, unlike other works that analyze the convergence rate of optimization schemes such as gradient descent. To establish an upper bound of the expected population mean-square prediction error for a convolutional filter, the authors adopt a localized empirical process approach (van de Geer, 2000) instead of attempting a closed-form solution due to difficulties analyzing the distribution properties of the input. They follow standard information-theoretic approaches to establish a lower bound on the expected population mean error. With these results in hand, they consider a network with a single hidden convolutional layer and establish an upper bound that depends on parameter count of the network as well as filter stride. The lower bound for the 1-layer network is shown to scale on the parameter count. A small set of experiments comparing a convolutional filter to a similar fully connected parameterization show that error goes up as filter size increases for a stride of 1, which corresponds to the derived error bound. The theoretical bound in which stride is not a factor has empirical evidence in the experiment setting the stride equal to the filter size. The experiments in Figure 4 show that a 1-hidden layer CNN with similar number of parameters and stride=1 exhibits similar performance to a FNN, but shared parameters and increased stride widens the gap (CNN outperforms FNN). Overall, I think the analysis is a helpful step towards understanding how CNNs learn more compact representations and require fewer samples for learning. It would have been nice to see some further empirical evidence - for example changing the input dimension or the type of data. Explanation of the experiments in Figure 4 are unclear. First it is stated that filter size is fixed to m = 8. Then, further explanation states that filter size is chosen to be 4 and 8. Is this supposed to be the stride?

Reviewer 3



Sample complexity for a linear convolution network based regression model is provided. Extension of the provided theoretical analysis for nonlinear models is not straight forward. In that sense, I would not refer to it as the sample complexity for a convolution neural network. Nevertheless, the theoretical results are valuable, possibly a stepping stone for deriving sample complexity on deep convolution neural networks in the future. Here are some suggestions. It may benefit to explain the difference between population and empirical loss. The mathematics in the introduction section may be avoided, and put in some other section. What are the connections, if any, between the sample complexity and optimization algorithm such as stochastic gradient ? For deep convolution neural networks, sample complexity analysis, explicitly accounting for the details of an optimization approach, could be beneficial and of high practical value. In Figure 4, the range for the number of training data is too short. For experiments if not theory, deep convolution neural networks should also be considered in the comparison. ------------------------------------------ I have read the rebuttal. While keeping same scores, I am in the favor of accepting this paper. I have increased my score by one point after a discussion with the other reviewers.